# The Relationship Between Widespread Pollution Exposure and Oxidized Products of Nucleic Acids in Seminal Plasma and Urine in Males Attending a Fertility Center

**DOI:** 10.3390/ijerph17061880

**Published:** 2020-03-13

**Authors:** Diana Poli, Roberta Andreoli, Lucia Moscato, Giovanna Pelà, Giuseppe de Palma, Delia Cavallo, Marta Petyx, Giorgio Pelosi, Massimo Corradi, Matteo Goldoni

**Affiliations:** 1Department of Occupational and Environmental Medicine, Epidemiology and Hygiene, INAIL Research, Via Fontana Candida1, 00078 Monte Porzio Catone, Rome, Italy; d.poli@inail.it (D.P.); d.cavallo@inail.it (D.C.); m.petyx@inail.it (M.P.); 2Department of Medicine and Surgery, University of Parma, via A. Gramsci 14, 43126 Parma, Italy; roberta.andreoli@unipr.it (R.A.); giovanna.pela@unipr.it (G.P.); massimo.corradi@unipr.it (M.C.); 3Centre for Research in Toxicology (CERT), University of Parma, via A. Gramsci 14, 43126 Parma, Italy; 4Center of Reproductive Infertility (CIR), University Hospital of Parma, via A. Gramsci 14, 43126 Parma, Italy; lmoscato@ao.pr.it; 5University Hospital of Parma, via A. Gramsci 14, 43126 Parma, Italy; 6Department of Medicine, Surgery, Radiological Sciences, Public Health and Human Sciences Unit, University of Brescia, 25121 Brescia, Italy; giuseppe.depalma@unibs.it; 7Department of Chemistry, Life Sciences and Environmental Sustainability, University of Parma, Parco Area delle Scienze 11/a, 43124 Parma, Italy; giorgio.pelosi@unipr.it

**Keywords:** environmental contaminants, idiopathic male infertility, oxidized products of nucleic acids, multiple exposure

## Abstract

Background: In recent decades, there has been an increase in male infertility, and in many cases, the etiology remains unclear. Several studies relate male hypo-fertility to xenobiotic exposure, even if no data exist about multiple exposure at the environmental level. Methods: The study involved 86 males with diagnosis of idiopathic male infertility (IMI), and 46 controls with no alteration in sperm characteristics. Seminal plasma (SP) and urine samples were analyzed by liquid chromatography tandem mass spectrometry (LC-MS/MS) to quantify biomarkers of exposure (the main metabolites of benzene, toluene, 1,3-butadiene, 3-monochloropropanediol, styrene, and naphthol) and effect (oxidized products of nucleic acids).Results: Biomarker concentrations were similar in subjects with IMI and controls even if a stronger correlation between biomarkers of exposure and effects were observed in SP. Data show that, both in SP and urine, most metabolites were inter-correlated, indicating a simultaneous co-exposure to the selected substances at the environmental level. Principal component analysis showed in SP the clustering of mercapturic acids indicating a preferential metabolic pathway with Glutathione (GSH) depletion and, consequently, an increase of oxidative stress. This result was also confirmed by multivariable analysis through the development of explanatory models for oxidized products of nucleic acids. Conclusions: This study highlights how oxidative stress on the male reproductive tract can be associated with a different representation of metabolic pathways making the reproductive tract itself a target organ for different environmental pollutants. Our results demonstrate that SP is a suitable matrix to assess the exposure and evaluate the effects of reproductive toxicants in environmental/occupational medicine. The statistical approach proposed in this work represents a model appropriate to study the relationship between multiple exposure and effect, applicable even to a wider variety of chemicals.

## 1. Introduction

The time-dependent degradation in human semen quality has become an important public health issue [1,2]. According to evidence, the World Health Organization (WHO) has recently lowered the accepted values for normal semen parameters (count, motility, and morphology) [3]. This has raised new concerns about occupational and environmental exposure to pollutants or toxicants that might affect human fertility [4], even if the etiology remains often unknown [1,2,5].

Several studies relate male reproduction system disorders to chemical agents, but the relationship between exposure and effect may vary, due to the different mechanisms of toxicity and routes of exposure [6,7,8]. Acute exposures to toxicants that can induce changes in semen parameters are uncommon, while chronic exposures are hard to identify. However, it is known that the effects of chronic, low-dose exposures from sources such as environment, food, smoking habits, and drug use can impair spermatogenesis and male fertility. In fact, male reproductive health may be a pollution-sensitive target [4,9,10,11,12]. Human exposure to environmental pollutants is widespread and occurs through multiple routes with a deep impact on human populations (e.g., pesticides, endocrine disrupting chemicals) [6,13,14].

Additionally, the effect of smoking habits on male fertility is still controversial because of the presence of more than 4000 chemical compounds in cigarette smoke with a different degree of toxicity and genotoxicity [15,16]. This complex scenario shows how low-dose exposure to fertility-related chemicals is extremely various and difficult to trace, making it difficult to identify and validate suitable biomarkers of exposure at environmental concentrations. On the other hand, the identification of proper biomarkers of effect could provide information between chemical exposure and reproductive system disorders.

It is well known that oxidative stress induced by xenobiotics can affect the male reproductive tract and spermatogenesis, which results in the production of sperm of poor quality [17,18]. Therefore, oxidized products of nucleic acids measured in biological fluids, such as seminal plasma (SP) and urine could be suitable biomarkers of the effects of oxidative stress.

These biomarkers arise from different repair pathways and/or turnover of nucleic acids. 8-Oxo-7,8-dihydroguanine (8-oxoGua), 8-oxo-7,8-dihydroguanosine (8-oxoGuo), and 8-oxo-7,8-dihydro-2’-deoxyguanosine (8-oxodGuo) may be generated from the native forms, depending on the localization of the guanine residue (DNA, RNA, or the {2′-deoxy}ribonucleotide pool), the involved repair systems of DNA/RNA, and the nucleic acid turnover [19]. Urinary 8-oxoGua is the most abundant oxidation product, due to the low oxidation potential of guanine at the C-8 position and, at least in part, from the glycosylase activity of the base excision repair (BER) system on oxidized guanine residues of DNA [20,21].

The origin of 8-oxodGuo is thought to be due to DNA multiple repair systems [21,22,23], but potential confounding factors related to cell turnover and diet have been ruled out [24]. Finally, 8-oxoGuo in urine is related to RNA turnover and/or repair mechanisms not definitively established [25,26]. The simultaneous determination of 8-oxoGua, 8-oxoGuo, and 8-oxodGuo in biological fluids may help to define the proportion of oxidative damage in progress at the DNA/RNA level.

The goal of this work is to present and discuss the relationship between low doses of exposure to environmental contaminants and oxidized products of nucleic acids in males attending a fertility center. To study the biomarkers distribution and oxidative stress in the male reproductive tract, both urine and seminal plasma (SP) were analyzed. Finally, an approach based on a multivariate model accounting for the multiple exposure was applied.

## 2. Materials and Methods

### 2.1. Study Design

A total of 163 males with suspected infertility were enrolled at the Center of Medically Assisted Procreation (CMAP) of the “Ospedale Maggiore di Parma” Hospital according to the World Health Organization (WHO) guidelines [3]. Subjects were selected on the basis of the failure to achieve a clinical pregnancy after 12 months, or more, with regular unprotected sexual intercourse. Thirty-one patients with a positive andrological history of infertility were excluded from the study for known causes (e.g. varicocele, infections, ejaculatory and erectile dysfunction, and endocrine dysfunction). Eight-six had a subsequent diagnosis of idiopathic male infertility (IMI), and 46 were classified as controls, owing to no alterations in sperm characteristics, and were due to female counterpart.

Informed consent was obtained from each subject at the time of enrollment. Our local ethics committee (Parma, Comitato Etico Provinciale) approved the study and all of the subjects gave their written informed consent (Protocol No 36206, 10/01/2012).

Before entering the study, they were administered a detailed screening questionnaire covering their previous medical history, smoking habits, and working history. The characteristics of the study population are shown in Table 1. Tobacco smoke exposure was evaluated based on self-reported current smoking status. The number of cigarettes smoked per day and the numbers of years of smoking were recorded. The subjects who had stopped smoking at least 1 year before recruitment were defined “ex-smokers”.

### 2.2. Sample Collection and Analysis

Semen samples were produced by masturbation at the clinical andrology laboratory after a period of 48 to 72 hours of sexual abstinence. After liquefaction, samples were evaluated for sperm concentration and motility according to the WHO guidelines [3]. Morphology smears were scored using the WHO classification, sperm concentration was expressed as 10^6^ per milliliter of semen, whereas motility and morphology were expressed as percentage. According to WHO 2010 criteria, sperm parameters are considered normal when sperm concentration is ≥15 million spermatozoa per milliliter, total motility and progressive motility are ≥40% and ≥32%, respectively, and sperm morphology (percentage of normal forms) ≥4%.

Within two hours from ejaculate collection and its liquefaction, semen was fractionated, collecting the acellular component known as seminal plasma (SP). The cellular fraction, which in physiological conditions mainly corresponds to spermatozoa, was not available in this study.

One ml of urine and 300 µl of SP samples were collected, centrifuged for 5 min at 10,000 rpm and stored at −20 °C until analysis to measure the main metabolites of benzene, toluene, 1,3-butadiene, 3-MCPD, styrene, and naphthalene and oxidized products of nucleic acids (8-oxo-Gua, 8-oxo-Guo and 8-oxo-dGuo). 8-oxo-Gua, 8-oxo-Guo, and 8-oxo-dGuo are not affected by circadian rhythm and therefore, sampling time is not a critical factor for their urinary determination [27]. Cotinine, a metabolite of nicotine, was measured to assess the exposure to cigarette smoke and a cut-off the value of urinary cotinine (U-Cot) >30 µg/L was used to identify the current smokers [28].

All biomarkers (exposure and effect), except the main metabolite of 1,3-butadiene (1,2-dihydroxybutilmercapturic acid—DHBMA) and of toluene (S-benzylmercapturic acid —SBMA), were measured as previously described (See Table 2) [27,29,30,31,32,33] using liquid chromatography tandem mass spectrometry (LC-MS/MS) AB-Sciex API 4000 triple quadrupole mass spectrometer (Applied Biosystems, Thornhill, Canada) equipped with a Turbo Ion Spray interface for pneumatically assisted electrospray. Compared to what is previously published, the use of new instrumentation, API 4000 instead of API365, allowed us to calculate new limit of detections (LODs) for Naphthol glucuro (NG), sulpho- conjugates (NS), 4-vinylphenol glucuro (VPG), and sulpho- conjugates (VPS) equal to 0.4 and 0.2 μg/L, respectively.

DHBMA and SBMA were measured by applying the method described by Manini et al. 2006 [29] for the determination of metabolites of industrial chemicals. A stable isotope-labelled compound, DHBMA-d_7_, was used as internal standard (IS). Ionization was performed in negative ion mode and the detection was obtained in selected reaction mode (SRM) monitoring the transitions m/z 250→121 and m/z 257→119 for DHBMA and DHBMA-d_7_, respectively and transition m/z 252→123 for SBMA. Calibrations were performed in matrix, by spiking samples from non-exposed subjects with standard mixtures at five different concentration levels, 1–10 μg/L for SP and 30–300 μg/L for urine. Since a blank urine for DHBMA is not available, the limit of detection (LOD, S/N = 3) and precision were determined in water and compared with those obtained in matrix using the internal standards. LOD was 0.01 μg/L, while intra- and inter-day precisions, calculated at two concentration levels (corresponding to the lowest and the highest values of the calibration range), were always lower than 5%.

Urinary concentrations of biomarkers of exposure and effect were expressed as a function of creatinine, measured by the Jaffe method [34]. All urine samples were considered acceptable because the creatinine concentration was within the range of 0.3–3.0 g/L [35].

### 2.3. Statistical Analysis

LODs were calculated as the signal-to-noise ratio equal to 3. In urine samples, all metabolites were detected, while in SP <5% of them was below LODs. In these cases, for each undetectable metabolite, a value equal to LOD/2 was used. Normality of data distribution was assessed by Shapiro–Wilk test. Log-normally distributed variables were reported as geometric mean (GM) and geometric SD (GSD), non-normalizable variables were reported as median and interquartile range (IQR). For all the variables, IMI subjects and controls were compared to assess the differences in conventional clinical data, and biomarkers of exposure and effect. Appropriate comparison tests were used depending on the distribution (t-test on logarithms or Man–Whitney–Wilkoxon test). Chi-square test was used to compare categorical variables. Given the absence of outliers with an excessive influence on models with not-transformed or log-transformed variables, Pearson’s univariate correlation coefficient was calculated always as log values. To assess the overall relationship between biomarkers of multiple exposure effects, models of principal component analysis (PCA) were used with the following options: varimax rotation, communality always near 0.7, starting eigenvalues >1, overall explained variance well above 50%.

Finally, a multiple regression model was used, including all the relevant variables that showed significance at the univariate analysis and the scores of factors resulting from PCA (regression method). Only variables with a variance inflation factor (VIF) <2 were included to exclude multicollinearity. A *p* value of 0.05 was always considered significant. All of the statistical analyses were made using statistical software IBM SPSS Statistics 25 (IBM, Amork, NY).

## 3. Results

The characteristics of IMI subjects and controls were similar in terms of age, body mass index (BMI), failure to achieve a clinical pregnancy, smoking habit, consumption of alcohol, and solvent exposure (Table 1). Taking into account the sperm characteristics on which the diagnosis of IMI is based, SP number, motility, and morphology of sperm were statistically different between two groups. Sperm count (10^6^/ml) was 24.0 (4.0–47.0) in IMI and 40.0 (30.5–59.0) in controls, sperm motility (%) was 55 (26–75) in IMI and 68 (59–77) in controls and sperm abnormal forms (%) were 3 (1–4) in IMI and 4 (4–5) in controls. All these data are expressed as median (25th–75th percentile).

Biomarkers of exposure and oxidized products of Guanine (Gua), Guanosine (Guo), 2’-deoxyguanosine (dGuo) were detectable in all urine samples, while t,t-muconic acid (t,t-MA), mandelic acid (MA), phenylglyoxylic acid (PGA), and phenylhydroxyethylmercapturic acid (AFIEM) were not detectable in SP samples, despite the high sensitivity of LC-MS/MS. All biomarker concentrations showed no significant differences between groups in both SP and urine. Table 1 displays the results of all the subjects.

Considering the overall subjects, no correlation between levels of biomarkers of effect and semen parameters were observed in SP (data not shown), while 8-oxoGuo and 8-oxo-dGuo concentrations correlated with sperm motility with R^2^ = 0.24 as Pearson’s correlation coefficients. With regard to urine samples, mercapturic acids of 1,3-butadiene (DHBMA) and 3-MCPD (DHPMA) levels correlated negatively with sperm count (R^2^ = −0.24) and sperm abnormal forms (R^2^ = −0.30), respectively.

Some mercapturic acids (DHBMA, DHPMA, SBMA, VPmerc) and cotinine concentrations correlated itself in both matrices (See Appendix A), while oxidized products of nucleic acids showed no correlation.

Differences between biomarker concentrations in current smokers and no- or ex-smokers were further analyzed (Table 3). In urine, biomarker concentrations were consistent with those reported in the literature for environmental or occupationally exposed non-smokers. Biomarkers of exposure levels were generally higher in smokers, especially those related to chemicals known to be present in cigarette smoke (e.g. benzene, 1,3-butadiene, styrene, naphthol). However, among biomarkers of effect, only 8-oxodGuo concentration was significantly higher in SP sample from smokers. Positive correlations between biomarkers (exposure and effect) versus cotinine were observed in SP, while in urine only biomarkers of effect showed positive correlations (Table 4).

Based on a positive self-reported job history for solvent exposure, no significant differences of effect were found through biomarkers both in SP and urine.

Biomarkers of exposure correlated each other in both SP and urine, with a high number of Pearson’s coefficients >0.5 (See Appendix A), confirming a multiple concurrent exposure to several contaminants at low concentrations.

Only in the case of styrene, it was possible to measure the same metabolites (VPtot and VPmerc), both in SP and urine, since 4-vinylphenol glucuro- and sulpho- conjugates (VPtot), MA, PGA, and AFIEM were not detectable in SP samples. The ratio between VPtot (the sum of 4-vinylphenol-glucuro- and sulpho-conjugates) and 4-vinylphenylmercapturic acid (VPmerc) was about 50 times higher in urine than in SP (see Appendix A).

PCA performed on inter-correlated biomarkers of exposure distinguished two factors and three factors with eigenvalues >1, in SP and urine, respectively (Table 5). With regard to SP, the first factor grouped mercapturic acids (DHBMA, DHPMA, SPMA, SBMA, and VPmerc), while the second grouped glucuro- and sulpho- conjugate forms (VPtot and NOHtot). The total explained variance of the model was 77%. On the other hand, in urine, the same cluster of biomarkers was not found: mercapturic acids were distributed in all three factors and the metabolites related to cigarette smoking were mostly in factors 2 and 3. The total explained variance of the model was 69%.

In SP, factor 1 was not significantly correlated with smoking status, while there was a significant but weak correlation between factor 2 and smoking status (R = 0.29, *p* < 0.01). In urine, all three factors were related to smoking habits, with a weak correlation for both factor 1 and 2 (R = 0.20, *p* < 0.05 and R = 0.25, *p* < 0.01, respectively) and high correlation for factor 3 (R = 0.58, *p* < 0.001). By using cotinine values instead of smoking status (yes/no), results were overlapping, despite the limitations due to the bimodal distribution of cotinine values. In fact, in SP, the correlation between factor 2 and cotinine remained weak (*r* = 0.32, *p* = 0.001), while in urine there was only a strong correlation between factor 3 and cotinine (r = 0.61, *p* < 0.001).

Looking at the relationship between biomarkers of exposure and effect, generally a higher number of significant correlations were observed in SP than in urine, especially in the case of 8-oxodGuo (Table 6).

This result was also confirmed by multivariable analysis through the development of explanatory models for 8-oxoGuo and 8-oxodGuo (Table 7 and Table 8). With regard to SP, sperm motility, smoking habit (Yes/No), solvent exposure (Yes/No), factors 1 and 2 resulting from principal component analysis were used as predictors. The same analysis was repeated by including the volume of seminal fluid as a predictor to evaluate its possible weight as a dilution factor. In the case of urine, the multivariate analysis was carried out considering as predictors sperm motility, solvent exposure (Yes/No), creatinine concentration as urine dilution factor, and factors 1, 2, and 3 resulting from principal component analysis. Smoking habit (Yes/No) was excluded due to its collinearity with the factor 3.

In SP, the explanatory model for both 8-oxoGuo and 8-oxodGuo showed the strongest significant associations with factor 1 and factor 2, while the volume of seminal fluid was significant only in the case of 8-oxoGuo, marginally affecting the significance of the factors. On the other hand, in urine samples, creatinine showed the strongest association with both biomarkers of effect. Factors 1 and 3 resulting from PCA were significant only in the case of 8-oxoGuo. Regarding 8-oxoGua, multivariable analysis did not show any statistically significant factor (data not shown).

## 4. Discussion

Exposure to environmental pollutants is widespread and occurs through several routes. Since humans are exposed to multiple combinations of compounds with anti-androgenic effects, it is important to assess the overall effect of multiple exposure. Many studies have demonstrated the anti-androgenic effect on adults exposed to several chemical classes: e.g. pesticides [13], perfluorinated chemicals [39], brominated flame retardants [40], phthalates [41,42].

In this study, we selected chemicals known to cause oxidative stress and with relevance from environmental and occupational point of view, and with suspected male reproductive toxicity (Table 2). Specifically, we focused our attention on benzene, toluene, styrene, 1,3-butadiene, naphthalene. These are known and ubiquitous air pollution components both in industrial and urban areas. They are emitted in the atmosphere by vehicular traffic, fires, industrial activities, environmental tobacco smoke in indoor environments [43,44,45]. Finally, we selected 3-MCPD, a compound with a documented effect on rat male fertility [46], as a representative of food contamination, being food an important source of chemical exposure. Additionally, we tested the relationship between 3-MCPD and the other compounds. This pattern of substances does not cover all possible contaminants with reproductive toxicity, however it can be considered a starting point to create a multivariate model to explain multiple exposure and to be enriched in the nearest future with other exposure data.

Biological monitoring is widely applied in environmental/occupational toxicology due to its remarkable utility on providing an efficient and effective means of measuring human exposure to chemical substances. It is mainly aimed at defining the existence of former exposure and quantifying the level of internal dose [29,47]. Blood and urine are by far the most used matrices, as they provide pertinent information about the biotransformation of xenobiotics in the body. In the study, we focused our attention on urine analysis, because the exposure evaluation to the substances selected in this study are generally performed on urine. On the other hand, SP analysis may provide specific information on the male reproductive system and could not reflect the overall exposure [48]. Several studies have highlighted that oxidative stress in the male reproductive tract is related to toxic agent exposure and it is implicated in low sperm quality (motility, viability, and increasing sperm morphological defects) [6,7,9,11,12]. A higher xenobiotic exposure (e.g., acute exposure) could also impair sperm membrane and affect spermatozoa triggering a cascade effect (e.g., increased cell permeability, enzyme inactivation, protein oxidations, and DNA damage).

In this case, the DNA damage to sperm due to xenobiotics in SP from widespread pollution, is difficult to explain. In fact, the low exposure and the relatively short period of contact between sperm and SP (<2 hours) makes it unlikely that oxidized products of nucleic acids are the result of repair enzyme activity. Additionally, the DNA within sperm is in a condensed form of heterochromatin, which is not readily available for attack by pollutants in SP. Alternatively, the pollutants could have crossed the blood–testis barrier (BTB) and affected the actively dividing spermatogonia resulting in damaged DNA. The plausibility of this concept is low in conjunction with the fact that (1) the arterial blood flow to the testis is rather low (about 9.6 mL/min out of a cardiac output of ~4.7 L/min) [49,50] and (2) the BTB is one of the tightest blood-tissue barriers known to exist in mammalian tissues, even more difficult to cross than the blood brain barrier [51,52,53]. In fact, one of the main role of the BTB is to prevent small molecules, and hence also harmful contaminants, from passing through the paracellular space [54]. On the other hand, despite being one of the strongest blood–tissue barriers, recent studies reported adverse effects of xenobiotics, such as bisphenol A and monobutyl phthalate, on Sertoli cell BTB [51,52]. To clarify this aspect, further investigation should be performed, particularly on spermatozoa, that unfortunately were not available to this study. In fact, several secreting epithelia and glands differentially concur to the production of SP. In particular, vesicular glands and prostate provide the main contribution releasing about 65% and 25% of SP, respectively, while testes and epididymes supply solely 2–5% of SP. As a mixture of various secretions from male accessory sex glands contains several biochemical compounds, also deriving from chemical exposure [55]. Additionally, free purines and ribonucleosides in blood (0.4–0.6 μM concentration) [56], after being incorporated into the secretions of the seminal vesicles and/or prostate, could become a possible target of oxidative stress in SP.

These observations led us to measure metabolites of xenobiotics in SP with the aim to relate biomarkers of oxidative stress at the male reproductive tract level and corresponding biomarkers of environmental exposure at low doses. Since it is known that lifestyle factors may potentially affect human fertility, several items were included in the questionnaire administered during recruitment. Characteristics of the groups were similar with the exception of sperm parameters on which the diagnosis of infertility is based. Overweight status and age, which several studies propose as risk factors affecting semen quality [57,58,59], were not different between IMI subjects and controls, probably because both parameters had relatively low dispersion. These parameters were not significantly correlated with both biomarkers of exposure and effect (data not shown), and therefore, they were not further used as possible confounding factors.

With regard to SP samples, all biomarkers were measurable, except t,t-MA, MA, PGA, and AFIEM that were measurable only in urine. Statistical analysis showed no significant differences between groups in SP and urine, maybe due to the recruitment of men from the general population, from the same geographical area, and with homogeneous characteristics of environmental/occupational exposure (Table 1). Overall, IMI diagnosis was not related to variations in biomarker concentration in both matrices. Therefore, further analysis was performed considering the overall subjects.

Correlations between biomarkers and male infertility parameters were not significant with the exception of mercapturic acids of 1,3-butadiene (DHBMA) and 3-MCPD (DHPMA) in urine samples, which were negatively correlated with sperm count and sperm abnormal forms. These data, although not conclusive due the low sample size, seem to suggest a weak toxic action performed by some xenobiotics on the male reproductive tract also at environmental concentrations, although the lack of significant correlations in SP makes this effect not specific and not related to spermatogenesis process. [48,55]. On the other hand, the relatively weak correlations between the same biomarker in SP and urine suggest that the xenobiotic metabolism and oxidative stress are influenced by the local target tissue concentration and metabolic pathways. This hypothesis is further reinforced looking at the Vinylphenol glucuro- and sulpho-conjugates (VPtot) and Vinylphenol mercapturic acid ratio and the fact that metabolites of benzene and two out of three metabolites of styrene were not measurable in SP. Therefore, SP could be considered a promising matrix to measure toxicants or their metabolites at the male reproductive tract level, when reproductive effects are expected.

Due to the low environmental/occupational exposure of the subjects, smoking habits could have an influence on SP and urine biomarker concentration. Cigarette smoke is a probable risk factor for male infertility because of the presence of a large amount of oxidants and toxic compounds [15,16]. Some chemicals selected in this study are metabolites of substances present in cigarette smoke: benzene, 1,3-butadiene, nicotine, toluene, and styrene and their presence can affect the indoor air pollution [44,60]. In this work, the smoker/non-smoker status was based on cotinine cutoff at >30 ug/L in urine. Generally, smokers show urine cotinine concentrations much higher than non-smokers: 512.9 (7.3) µg/g creat versus 2.5 (4.4) (µg/g creat as geometric mean (GSD). Since there was no overlap between cotinine concentrations in the two groups, the smoker/non-smoker status may be discriminated by cotinine. The cotinine variables were completely bimodal both in urine and SP, and their use as continuous variables may create statistic artifacts. Significant differences in several biomarkers of exposure between smokers and non-smokers were detected in both biological matrices, but only SP samples contained significantly higher levels of 8oxodGuo in smokers, in agreement with literature [61,62]. Additionally, 8oxoGuo and 8-oxodGuo concentrations in SP showed positive correlations with cotinine. These results confirm once again how SP may be representative of local toxic effects, like oxidative stress [55]. Furthermore, considering the working history of the subjects, oxidized products of nucleic acids were not influenced by solvent exposure while only some biomarkers of exposure showed significant differences. With the limitation that it was self-reported non-quantitative information, occupational exposure seems to have no effect on the concentration of biomarkers of exposure.

Generally, it was not possible to distinguish a priori the sources of exposure because the selected chemical compounds are ubiquitous at very low concentrations. Therefore, we focused our attention to test if there was a relationship among them. In fact, the main goal of the PCA model was to find patterns of substances that could be independently related to biomarkers of effect. Given the results, sperm motility, smoking condition, and exposure solvent were included in the multivariate models.

The data about correlations between biomarkers of exposure and effects show that: (1) both in SP and urine, most metabolites were inter-correlated, indicating a simultaneous co-exposure to the selected substances at the environmental level; (2) there were several correlations between metabolites and oxidized bases both in SP and urine, particularly strong for 8-oxodGuo in SP and 8-oxoGuo in urine.

At this level, it would be difficult to weigh the contribution of each metabolite in determining the effect also in multivariate models due to the high risk of multi-collinearity. On the other hand, PCA analysis could be extremely useful to sum up the contribution of inter-correlated substances, particularly when the extrapolated factors from PCA itself may explain a high part of multivariate variance. Furthermore, PCA could create clusters of substances with partially common origin, where the weight of each of them could be assessed by the relative correlation coefficient with the factor. Therefore, such an approach is very promising to test the multiple exposure also with a number of toxic substances higher that the compound models tested in this study.

PCA showed in SP the clustering of mercapturic acids (Table 5). This result and the fact that the ratio between VPmerc and VPtot was higher in SP than in urine indicates a preferential testicular metabolic pathway with GSH depletion and, consequently, an increase of oxidative stress. In fact, this antioxidant, which is a co-factor for glutathione peroxidase, reacts directly with reactive oxygen species (ROS) as a scavenger, and protects primarily against products associated with lipid peroxidation. Glutathione level, when reduced in seminal fluid, is associated with poor semen quality and oligozoospermia [63]. Although the use of antioxidants as a therapeutic option is useful in male infertility, caution to this approach is required to avoid reductive stress, which may become as dangerous as oxidative stress [64,65]. A balanced ROS concentration in the healthy male reproductive tract is essential to the acquisition of fertilizing ability [66].

The high vulnerability of the male reproductive tract to oxidative stress caused by chemicals was demonstrated by the higher number of correlations between biomarker of exposure and effect in SP, especially in the case of 8-oxodGuo (Table 7). This result was also confirmed by multivariable analysis through the development of explanatory models for 8-oxoGuo and 8-oxodGuo (Table 7 and Table 8). The absence of a relationship between exposure and 8-oxoGua concentration is probably due to the highest inter- and intra individual variability of 8-oxo-Gua among biomarkers of nucleic acid oxidation investigated [23,27]. Statistical analysis showed in SP a direct relationship between exposure (factor 1 and factor 2 from PCA) and effect for both oxidized bases. Sample dilution alone was not able to explain 8-oxoGuo and 8-oxodGuo concentrations and the inclusion of SP volume as a predictor did not affect the result of multivariable analysis. Only factor 2 showed a weak relationship with smoking status, indicating that in SP, the contribution of cigarette smoke cannot be clearly distinguished from other pollutant sources.

On the other hand, in urine, two clusters of exposure, specifically factor 1, which contained some mercapturic forms, and factor 3, related to smoking condition, were significant predictors only for 8-oxoGuo. Local and systemic effects on oxidized nucleic acids are therefore not explainable by the same processes and by the same cluster of toxicant metabolites. In this case, the main contribution of cigarette smoke is clearly identifiable in factor 3.

## 5. Conclusions

Our work confirms the vulnerability of the male reproductive tract to xenobiotic exposure even at low concentration. This is mainly due to the local biotransformation of the toxic substances through the activation of secondary metabolic pathways (mercapturate pathway). The clustering of mercapturic acid supports the idea that GSH depletion might lead to an increased oxidative stress, making the male reproductive tract a target organ more susceptible to chemicals. Our overall data clearly demonstrate that SP and urine provide different information, confirming the possibility of using SP to assess exposure to reproductive toxicants, both in environmental and occupational medicine. Additional research is certainly needed, in particular by increasing the sample size and performing lipid peroxidation assessment in sperm to confirm our findings and to clarify the model of action of these chemicals.

However, our statistical approach has the features to be adopted as a model to study the relationship between multiple exposure and effect even with a higher number of toxicants.

## Figures and Tables

**Table 1 ijerph-17-01880-t001:** Main characteristics of the study groups.

	Subjects (*n* = 132)	
Age (years)	36.7 (5.5) ^1^	
BMI	25.8 (3.2) ^1^	
Smokers	22.5 (42.6%) ^1^	
Pack/years	5.0 (1.5–8.9) ^2^	
Consumption of alcohol (Yes)	8.5 (13.3%) ^1^	
Solvent exposure (Yes)	35.4 (51.9%) ^1^	
Failure to achieve a clinical Pregnancy (months)	24 (12–36) ^2^	
**Biomarkers**	**Seminal Plasma (** **μ** **g/l)**	**Urine (** **μ** **g/g creat)**
DHBMA	3.6 (2.75) ^3^	131.8 (1.7) ^3^
DHPMA	13.8 (2.4) ^3^	588.8 (1.4) ^3^
SPMA	0.13 (2.63) ^3^	0.30 (1.89) ^3^
SBMA	0.27 (1.86) ^3^	5.06 (2.36) ^3^
VPmerc	2.51 (4.97) ^3^	4.16 (3.02) ^3^
VPtot	2.75 (2.76) ^3^	190.0 (2.9) ^3^
AFIEM	n.d.	3.27 (1.99) ^3^
NOHtot	2.29 (2.51) ^3^	24.5 (2.07) ^3^
t,t-MA	n.d.	68.9 (2.4) ^3^
MA	n.d.	191 (1831) ^3^
PGA	n.d.	201 (2157) ^3^
Cot	6.72 (13.6) ^3^	15.5 (20.4) ^3^
8oxoGua	28.8 (15.8) ^3^	8.83 (2.84) ^3^
8oxoGuo	2.22 (1.99) ^3^	6.92 (1.54) ^3^
8oxodGuo	1.72 (2.40) ^3^	3.47 (1.96) ^3^

^1^ Mean (Standard Deviation); ^2^ Median (25th–75th percentile); ^3^ geometric mean (Geometric Standard Deviation); n.d. = non-detectable; DHBMA: 1,2-dihydroxybutilmercapturic acid; DHPMA: 2,3-dihydroxypropylmercapturic acid; SPMA: phenylmercapturic acids; SBMA: S-benzylmercapturic acid; VPmerc: 4-vinylphenylmercapturic acid; VPtot: 4-vinylphenol glucuro- (VPG) and sulpho- conjugates (VPS); AFIEM: phenylhydroxyethylmercapturic acid; NOHtot: Naphthol glucuro- (NG) and sulpho- conjugates (NS); t,t,-MA: t,t-muconic acid; MA: mandelic acid; PGA: phenylglyoxylic acid; Cot: Cotinine: 8oxoGua: 8-oxo-7, 8-dihydroguanine; 8oxoGuo: 8-oxo-7, 8-dihydroguanosine; 8-oxodGuo: 8-oxo-7, 8-dihydro-2’-deoxyguanosine.

**Table 2 ijerph-17-01880-t002:** Parent compounds and the main metabolites investigated.

Parent Compound	Metabolites Investigated	LOD (µg/L)	Method of Analysis
Benzene	t,t-muconic acid (t,t-MA) phenylmercapturic acids (SPMA)	2.5 0.01	Manini et al., 2006 [29] Manini et al., 2008 [30]
Toluene	S-benzylmercapturic acid (IMI)	0.01	Based on Manini et al., 2006 [29]
1,3-Butadiene	1,2-dihydroxybutilmercapturic acid (DHBMA)	0.1	Based on Manini et al., 2006 [29]
Styrene	mandelic acid (MA) phenylglyoxylic acid (PGA) phenylhydroxyethylmercapturic acid (AFIEM) 4-vinylphenylmercapturic acid (VPmerc) 4-vinylphenol glucuro- and sulpho- conjugates (VPG and VPS = VPtot)	10 10 0.7 0.7 0.4/0.2 *	Manini et al., 2006 [29]
Naphthalene	Naphthol glucuro- and sulpho- conjugates (NG and NS=NOHtot)	0.4/0.2 *	Andreoli et al., 1999 [31]
3-monochloropropanediol (3-MCPD)	2,3-dihydroxypropylmercapturic acid (DHPMA)	2.2	Andreoli et al., 2015 [32]
Nicotine	Cotinine (Cot)	0.2	De Palma et al., 2012 [33]
DNA and RNA	8-oxo-7, 8-dihydroguanine (8-oxoGua) 8-oxo-7, 8-dihydroguanosine (8-oxoGuo) 8-oxo-7, 8-dihydro-2’-deoxyguanosine (8-oxodGuo)	1.0 0.3 0.2	Andreoli et al., 2010 [27]

* Compared to what previously published, the use of new instrumentation, API 4000 instead of API365, allowed to calculate new limit of detections (LODs) for SPMA, SBMA, NG, NS, VPG, and VPS.

**Table 3 ijerph-17-01880-t003:** Biomarkers of exposure and oxidized products of nucleic acids from smokers (*n* = 29) and non-smokers (*n* = 58).

	Seminal Plasma (μg/L)		Urine (μg/g creat)		
Biomarkers	Smokers	Non-Smokers	Smokers	Non-Smokers	Literature Data
DHBMA	4.0 (3.0)	2.7 (2.4)	167.3 (1.7) **	112.0 (1.7) **	187.6 (68.2–344.0) μg/l [36] ^§^
DHPMA	16.7 (2.5) *	11.3 (2.1) *	610 (3.2)	575 (2.1)	296 (233–388) μg/g creat [32] ^§^
MA	0.31 (1.9)	0.24 (1.8)	0.39 (1.92) **	0.24 (1.7) **	0.20 (1.63) μg/g creat [32] ^§^
VPmerc	2.8 (5.3)	2.4 (6.0)	5.5 (2.6) *	3.3 (3.1) *	110–150 μg/g creat [37] ^§§^
VPtot	3.6 (2.3) *	2.2 (2.7) *	325.1 (3.0) **	140.4 (2.6) **	3.97 mg/g creat [37] ^§§^
AFIEM	n.d.	n.d.	3.30 (2.3)	3.12 (1.9)	0.3–1.4 μg/g creat [38] ^§^
NOHtot	2.5 (3.2)	2.1 (2.2)	41.2 (1.8) **	17.5 (1.9) **	155 (32–503) μg/g creat [31] ^§§^
t,t-MA	n.d.	n.d.	116.9 (2.5) **	49.6 (2.2) **	48.1 (2.47) μg/g creat [32]^§^
MA	n.d.	n.d.	283 (1.7) **	164 (1.6) **	146 mg/g creat [37] ^§§^
PGA	n.d.	n.d.	241 (2.4) **	165 (2.1) **	83.4 mg/g creat [37] ^§§^
Cot	73.6 (6.2) **	1.4 (3.1) **	512.9 (7.3) **	2.5 (4.4) **	1.78 (2.66) μg/g creat [32] ^§^
8oxoGua	30.2 (1.5)	31.6 (1.8)	7.3 (2.8)	8.8 (3.1)	17.8 (2.66) μg/g creat [32] ^§^
8oxoGuo	2.4 (1.9)	1.9 (1.8)	7.0 (1.4)	6.75 (1.6)	9.43 (1.83) μg/g creat [32] ^§^
8oxodGuo	2.3 (2.1) *	1.4 (2.3) *	3.52 (1.9)	3.3 (1.0)	3.95 (1.47) μg/g creat [32] ^§^

* *p* <0.05; ** *p* < 0.01; Geometric mean (Geometric Standard Deviation); n.d. = non-detectable; ^§^ Non-smokers, general population; ^§§^ Occupationally Exposed.

**Table 4 ijerph-17-01880-t004:** Pearson’s correlation coefficients between biomarkers vs. cotinine.

	Seminal Plasma	Urine
Biomarkers	Cotinine	Cotinine
DHBMA	0.407 **	0.350 **
DHPMA	0.409 **	N.S.
SPMA	0.208 *	0.446 **
SBMA	0.313 **	N.S.
VPtot	0.470 **	0.466 **
NOHtot	0.300 **	0.602 **
t,t-MA	n.d.	0.444 **
MA	n.d.	0.600 **
PGA	n.d.	0.284 **
8oxoGuo	0.264 **	N.S.
8oxodGuo	0.450 **	N.S.

R * = *p* < 0.05; R ** = *p* < 0.01; N.S. = not significant (*p* > 0.05); n.d. = non-detectable.

**Table 5 ijerph-17-01880-t005:** Principal component analysis.

	**Seminal Plasma**		**Urine**		
	Factor 1	Factor 2	Factor 1	Factor 2	Factor 3
DHBMA	0.911	-	0.845	-	-
DHPMA	0.766	-	0.809	-	-
SPMA	0.836	-	-	-	0.493
SBMA	0.730	-	-	0.687	-
VPmerc	0.792	-	0.883	-	-
VPtot	-	0.856	-	-	0.873
NOHtot	-	0.860	-	-	0.745
t,t-MA	-	-	-	0.597	-
MA	-	-	-		0.593
PGA	-	-	-	0.809	-
AFIEM	-	-	-	0.751	-

**Table 6 ijerph-17-01880-t006:** Pearson’s correlation coefficients between biomarkers of exposure vs. biomarkers of effect.

	Seminal Plasma			Urine		
Biomarkers	8-oxoGua	8-oxoGuo	8-oxodGuo	8-oxoGua	8-oxoGuo	8-oxodGuo
DHBMA	−0.224 *	0.4 **	0.48 **	N.S.	0.61 **	0.31 **
DHPMA	N.S.	0.35 **	0.44 **	N.S.	0.64 **	0.52 **
SPMA	−0.215 *	0.38 **	0.47 **	N.S.	0.33 **	N.S.
SBMA	N.S.	0.42 **	0.51**	N.S.	N.S.	N.S.
VPmerc	N.S.	N.S.	0.3 *	N.S.	0.43 **	N.S.
VPtot	N.S.	N.S.	0.26 *	N.S.	N.S.	N.S.
AFIEM^§^	-	-	-	N.S.	0.403 **	0.434 **
NOHtot	N.S.	N.S.	0.32 **	N.S.	0.31 **	N.S.
t,t-MA^§^	-	-	-	N.S.	0.279 **	0.274 **
MA^§^	-	-	-	N.S.	0.473 **	0.361 **
PGA^§^	-	-	-	N.S.	N.S.	0.312 **

R * = *p* < 0.05; R ** = *p* < 0.01; N.S. = not significant (*p* > 0.05); ^§^ non-detectable in SP.

**Table 7 ijerph-17-01880-t007:** Multivariable analysis. Explanatory models for 8-oxodGuo using clinical variables, smoking/solvent exposure, and factors calculated using principal component analysis as predictors.

Coefficients	8-oxodGuo					
	SP		SP		Urine	
	β	*p*	β	*p*	β	*p*
Constant		<0.011		<0.01		<0.01
Sperm motility (10^6^/ml)	1e-4	0.733	-0.001	0.553	-0.001	0.498
Smokers (Yes/No)	0.07	0.332	0.06	0.334	-	-
Solvent exposure (Yes/No)	0.07	0.283	0.07	0.292	0.03	0.585
Factor 1	0.18	<0.001	0.15	<0.001	0.06	0.110
Factor 2	0.11	<0.01	0.09	<0.01	0.05	0.153
Factor 3	-	-	-	-	0.04	0.162
Semen volume	-	-	-0.06	<0.01	-	-
Log creatinine	-	-	-	-	0.51	0.013
	R^2^ = 0.41	*p* < 0.01	R^2^ = 0.41	*p* < 0.01	R^2^ = 0.24	*p* < 0.05

**Table 8 ijerph-17-01880-t008:** Multivariable analysis. Explanatory models for 8-oxoGuo using clinical variables, smoking/solvent exposure, and factors calculated using principal component analysis as predictors.

	8-oxoGuo					
	SP		SP		Urine	
	β	*p*	β	*p*	β	*p*
Constant		<0.001		<0.001		<0.001
Sperm motility (10^6^/ml)	−0.001	0.301		0.272	0.00	0.734
Smokers (Yes/No)	−0.03	0.632		0.643	-	-
Solvent exposure (Yes/No)	0.06	0.301		0.242	−0.50	0.136
Factor 1	0.122	<0.001		<0.001	0.07	0.002
Factor 2	0.085	0.002		0.005	0.01	0.830
Factor 3	-	-	-	-	0.05	0.007
Semen volume	-	-		0.402	-	-
Log creatinine	-	-	-	-	0.45	<0.001
	R^2^ = 0.34	*p* < 0.01	R^2^ = 0.39	*p* < 0.01	R^2^ = 0.31	*p* < 0.01

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
