# Peer review of "The Relationship Between Widespread Pollution Exposure and Oxidized Products of Nucleic Acids in Seminal Plasma and Urine in Males Attending a Fertility Center"

_ijerph, 2020, doi:10.3390/ijerph17061880_

Round 1

Reviewer 1 Report

Overall: Interesting paper however the conceptual framework is not clearly laid out and the analysis, though extensive does not follow a logical and consistent approach.

Major:

  1. This paper needs to make a stronger toxicological case for why these biomarkers of exposure were chosen in relation to semen health. Also stronger explanation of what volatiles are typically found in cigarette smoke.
  2. In the sample collection and analysis section page 4 the LOD or LOQ for the analytes referenced to other papers should still be listed.
  3. In the Statistical Analysis section not explanation was given for how analytes < LOD/LOQ were handled (LOD/2 etc). In the results, some indication of % of each analyte < LOD/LOQ should be reported.
  4. Given that the stated goal of this paper is “to present and discuss the relationship between low doses of exposure to environmental contaminants and oxidized products of nucleic acids in males attending a fertility center” I strongly believe that this paper needs to better differentiate the sources of the biomarkers of exposure: cigarette smoke, environmental exposures, occupational exposures.

2a. If you have cotinine measures, why are multivariable models simply using smoker yes/no?   Cotinine would avoid misclassification bias of self-report.

2b. All that is mentioned on occupational exposure is pg 7 line 197-200: “a positive self-reported job history for solvent exposure, some biomarkers of effect showed higher concentrations in both SP and urine”.  This would be a useful table, provided you controlled /stratified by smoking/cotinine.    

2c. You mention in title Environmental Contaminant Exposure, but do not address this in the text.  Could location of home or work be a proxy for traffic or heavy industrial exposure in the community?  What about commuting via car, bus etc. Mobile sources are an important contributor to many of these parent compounds.

  1. The identification of subjects and controls does not seem useful in relation to any of your main outcomes: the biomarkers of DNA damage 8oxo (Table 3 at end). So I don’t see the benefit of giving us the data in Table 2 by Cases and Controls……would suggest you simplify and put all subjects together (no difference in Table 2 or 3 except sperm count, motility and form which are not your outcomes!!). So then you end up with a simple demographics table without columns for subjects and controls (Table 2) where you could also include the 8oxo results for SP and urine (drop Table 3).
  2. Perhaps you can start off addressing occupational exposure as it does not seem very important……it could be modeled on Table 5 (see below). Since now all you say about occupational exposures is 1 long sentence on pa 7 line 197-200….was this solvent exposure ever or currently? If you have current data I would suggest you do comparison with that.
  3. A modified table 5 for plasma and urine levels of biomarkers of exposure (either drop or separate out and label biomarkers of effect …if include, include all 8-oxo) and include a column for all subjects (unless you do above with occ exposure table ).  This would be helpful when you get to discussion and need to compare biomarker levels found in your subjects vs other studies of environmental or occupational exposures.  Note: in pg 4 line 139-141 you say urinary concentrations were expressed as a function of creatinine but labels here are only ug/l not ug/g creatinine).
    1. Not clear if this “smoker/non-smoker” status was based on cotinine cutoff of > 30 ug/L page 4 Line 121 or self-reported status as listed in Table 2 (described page 3 line 101-104).
  4. In text you say lognormally distributed variables were reported as GM and GSD and that is shown in Table 3 and 4 but then you use pearson’s correlation to examine the relationship of biomarkers in SP and urine…..if these were on log values please specify, if not, they should be! The correlation of biomarkers of exposure in urine and SP and between each other can be discussed in text, I don’t really think you need tables as this is not surprising results (or could put in supplemental tables (Table 4, 7a, 7b) .
  5. Likewise, Figure 1 isn’t really necessary, since you have info in text.
  6. It seems to me that the next logical step is the PCA as a way of dealing with this correlation (Table 8). Then the question becomes, do these Factor combinations relate to cotinine levels? So Table 6 could be modified to show that (missing VPmerc and AFIEM and no need for 8oxo data) OR better still, you could simply model the factors relative to Cotinine as outcome.  Reminder that in pg 4 line 139-141 you say urinary concentrations were expressed as a function of creatinine but here in Table 6 you list ug/l not ug/g creatinine. …… what was PCA done on?
  7. Table 9. Is this on logs of outcomes and exposure biomarkers? If you want to do a uncontrolled analysis of outcomes with biomarkers…..why not use log of 8-oxo and the unlogged biomarker level as predictor….then Beta x100 is the percent change per unit increase in 8-oxo ?
    1. This table has no unites of 8-oxo
    2. Reminder that in pg 4 line 139-141 you say urinary concentrations were expressed as a function of creatinine but here in Table 9 you list ug/l not ug/g creatinine
  8. The Tables 10a, b are confusing….and what happened to 8-oxoGua?
    1. is the outcome logged or not?
    2. No case is made for why you include sperm motility or volume SF (isn’t this quite variable per subject collection, not really a biological factor?) or log creatinine (in pg 4 line 139-141 you say urinary concentrations were expressed as a function of creatinine)
    3. Why use smoker yes/no when you have cotinine which is a better marker of smoking status?

Editorial

Page 2. Line 53-54 add this to previous paragraph

Page 3 line 97 should be enrollment

Page 3 Line 98 specify organization that takes responsibility for ethical committee

Page 3 Line 101 should be Table 2

Page 3 Line 115 write out seminal plasma (SP) and define it since you also use the terms semen under sample collection analysis section or at least be consistent on which you use.

Table 2 footnote What is DS? Should be Median

Table 3 What is footnote DS? Do you mean GM for Geometric Mean and GSD for Geometric Standard Deviation??

Page 6 line 185 divide the words were further

Author Response

REVIEWER 1

Overall: Interesting paper however the conceptual framework is not clearly laid out and the analysis, though extensive does not follow a logical and consistent approach.

Major:

  1. This paper needs to make a stronger toxicological case for why these biomarkers of exposure were chosen in relation to semen health. Also stronger explanation of what volatiles are typically found in cigarette smoke.

REPLY: Our main goal was to investigate if ubiquitous environmental xenobiotics at low concentrations showed a relationship with oxidized products of nucleic acids not only in urine but also at male reproductive tract level.

Since exposure to environmental pollutants is widespread and occurs through multiple routes, we selected chemicals known to cause oxidative stress and with relevance from environmental and occupational point of view but less investigated with respect to male reproductive health disorder.

Benzene, toluene, styrene, 1,3-butadiene, naphthalene are well-known contaminants arising from vehicular traffic, indoor air mixed with environmental tobacco smoke, fires and in vitro and in vivo evidences suggest their reproductive toxicity. We decided to add a food contaminant, 3-MCPD, to test its relationship with the other compounds. Its effect on rat male fertility is also studied. This pattern of substances does not cover all possible contaminants with reproductive toxicity, but it could be considered a starting point to create a multivariate model of multiple exposure, to be enriched in the future with other exposure data.

Therefore, the following sentences have been deleted:

“In the present work, we have focused our attention on environmentally/occupationally relevant chemicals, where the effects on reproduction are mainly documented on animals. Specifically, we selected benzene, toluene, styrene, 1,3-butadiene, naphthalene as representatives of environmental and occupational pollutants and 3-MCPD as representative of food contaminants (Table 1).”

and these sentences have been added:

Discussion -  Line 262

“Exposure to environmental pollutants is widespread and occurs through multiple routes.”

Discussion -  Line 267

“In this study we selected chemicals known to cause oxidative stress and with relevance from environmental and occupational point of view, and with suspected male reproductive toxicity (Table 2). Specifically, we focused our attention on benzene, toluene, styrene, 1,3-butadiene, naphthalene. These are known and ubiquitous air pollution components both in industrial and urban areas. They are emitted in the atmosphere by vehicular traffic, fires, industrial activities, environmental tobacco smoke in indoor environments [47-49]. Finally, we selected 3-MCPD, a compound with a documented effect on rat male fertility [50], as a representative of food contamination, being food an important source of chemical exposure. Additionally, we tested the relationship between 3-MCPD and the other compounds. This pattern of substances does not cover all possible contaminants with reproductive toxicity, however it can be considered a starting point to create a multivariate model to explain multiple exposure and to be enriched in the nearest future with other exposure data.”

  1. In the sample collection and analysis section page 4 the LOD or LOQ for the analytes referenced to other papers should still be listed.

REPLY: To make text easier to understand LOD values were added directly in Table 1. Additionally, the following sentences have been added:

2.2. Sample Collection and Analysis – Line 132

“Compared to what was previously published, the use of new instrumentation, API 4000 instead of API365, allowed us to calculate new LODs for NG, NS, VPG and VPS equal to 0.4 μg/L and 0.2 μg/L, respectively.”

  1. In the Statistical Analysis section not explanation was given for how analytes < LOD/LOQ were handled (LOD/2 etc). In the results, some indication of % of each analyte < LOD/LOQ should be reported.

REPLY: The LOD was calculated as signal to noise ratio equal to 3. In urine samples, all metabolites were detected, while in SP < 5% of them were below LODs. In these cases, for each metabolite undetectable, a value equal to LOD/2 was used. These sentences have been added in the text.

2.3. Statistical Analysis – Line 152

“LODs were calculated as signal-to-noise ratio equal to 3. In urine samples, all metabolites were detected, while in SP < 5% of them was below LODs. In these cases, for each undetectable metabolite, a value equal to LOD/2 was used”.

  1. Given that the stated goal of this paper is “to present and discuss the relationship between low doses of exposure to environmental contaminants and oxidized products of nucleic acids in males attending a fertility center” I strongly believe that this paper needs to better differentiate the sources of the biomarkers of exposure: cigarette smoke, environmental exposures, occupational exposures.

REPLY: the main aim of the manuscript was not to differentiate the sources of biomarkers of exposure and characterize them, but to compare two possible matrices for biological monitoring and to propose a model to consider the multiple exposure at the background level. However, we did our best to consider all the concerns about this point proposed by the referee. Given the substances justified at the point 1, their low concentrations and the fact that we measured their metabolites, it not possible to distinguish a priori the sources of exposure: all of them are ubiquitous. The PCA model had just the main goal to find pattern of substances that could be independently related to biomarkers of effect. In the text, we made some hypothesis about the source of the factors based on their composition. To explain better this concept, the following statements have been added.

Discussion – Line 349

“Generally, it was not possible to distinguish a priori the sources of exposure because the selected chemical compounds are ubiquitous at very low concentrations. Therefore, we focused our attention to test if there was a relationship among them. In fact, the main goal of the PCA model was to find patterns of substances that could be independently related to biomarkers of effect.”

2a. If you have cotinine measures, why are multivariable models simply using smoker yes/no?   Cotinine would avoid misclassification bias of self-report.

REPLY: Smoker/non-smoker status was based on cotinine cutoff of > 30 ug/L in urine as reported in “2.2. Sample Collection and Analysis”. However, it is important to underlining that smoker subjects showed urine cotinine concentrations far more higher than non-smokers (see Table 5):  512.9 (7.3) mg/g creat versus 2.5 (4.4) (mg/g creat as geometric mean (GSD). As there was no overlap between cotinine concentrations between the groups, the smoker/non-smoker status may be based indifferently on cotinine concentrations or self-reported status (Yes or No). The cotinine variable in urine (but also in SP) was completely bimodal and its use as continuous variable may create statistic artifacts. However, we agree with the reviewer that this problem need to be more addressed. Therefore, the following sentences have been added.

Discussion section – Line 334

“In this work, the smoker/non-smoker status was based on cotinine cutoff at > 30 ug/L in urine. Generally, smokers show urine cotinine concentrations much higher than non-smokers:  512.9 (7.3) µg/g creat versus 2.5 (4.4) (µg/g creat as geometric mean (GSD). Since there was no overlap between cotinine concentrations in the two groups, the smoker/non-smoker status may be discriminated by cotinine. The cotinine variables were completely bimodal both in urine and SP, and their use as continuous variables may create statistic artifacts.”

2b. All that is mentioned on occupational exposure is pg 7 line 197-200: “a positive self-reported job history for solvent exposure, some biomarkers of effect showed higher concentrations in both SP and urine”.  This would be a useful table, provided you controlled /stratified by smoking/cotinine.    

REPLY: We are sorry, but after re-analyzing data, we noticed an error due to a wrong classification of solvent exposure. No biomarkers of exposure were significantly different considering solvent exposure. Therefore, we have not performed the table but we modified the text accordingly. However, we left this variable in the multivariate model. The following sentence has been added.

Results – Line 212

“Based on a positive self-reported job history for solvent exposure no significant differences of effect were found through biomarkers both in SP and urine.”

2c. You mention in title Environmental Contaminant Exposure, but do not address this in the text.  Could location of home or work be a proxy for traffic or heavy industrial exposure in the community?  What about commuting via car, bus etc. Mobile sources are an important contributor to many of these parent compounds.

REPLY: Almost all participants were Parma Citizens (>90%) and therefore it is very difficult to stratify them based on environmental exposure. We think that a slightly different title may result clearer.

New title:

“The Relationship Between Widespread Pollution Exposure and Oxidized Products of Nucleic acids in Seminal (…)”

  1. The identification of subjects and controls does not seem useful in relation to any of your main outcomes: the biomarkers of DNA damage 8oxo (Table 3 at end). So I don’t see the benefit of giving us the data in Table 2 by Cases and Controls……would suggest you simplify and put all subjects together (no difference in Table 2 or 3 except sperm count, motility and form which are not your outcomes!!). So then you end up with a simple demographics table without columns for subjects and controls (Table 2) where you could also include the 8oxo results for SP and urine (drop Table 3).

REPLY: According to reviewer’s suggestions, results in Table 2 and 3 were summarized in one each table (now Table 1) and all the subjects were analyzed together. Sperm characteristics on which the diagnosis of IMI is based, were reported directly in the text. The following sentences were added:

Results – Line 178

“Sperm count (106/ml) was 24.0 (4.0-47.0) in IMI and 40.0 (30.5-59.0) in controls, sperm motility (%) was 55 (26-75) in IMI and 68 (59-77) in controls and sperm abnormal forms (%) were 3 (1-4) in IMI and 4 (4-5) in controls. All these data are expressed as median (25th-75th percentile).

Results – Line 186

Results – Line 185

“Table 1 displays the results of all the subjects.”

  1. Perhaps you can start off addressing occupational exposure as it does not seem very important……it could be modeled on Table 5 (see below). Since now all you say about occupational exposures is 1 long sentence on pa 7 line 197-200….was this solvent exposure ever or currently? If you have current data I would suggest you do comparison with that.

REPLY: Unfortunately, the question in the subministered questionnaire was generic, and therefore it deals with “solvent exposure ever” and not currently. Therefore, we cannot perform any further comparison.

  1. A modified table 5 for plasma and urine levels of biomarkers of exposure (either drop or separate out and label biomarkers of effect …if include, include all 8-oxo) and include a column for all subjects (unless you do above with occ exposure table ).  This would be helpful when you get to discussion and need to compare biomarker levels found in your subjects vs other studies of environmental or occupational exposures.  Note: in pg 4 line 139-141 you say urinary concentrations were expressed as a function of creatinine but labels here are only ug/l not ug/g creatinine).

REPLY: Since we already put together all subjects (IMI subjects and controls) in Table 1, we prefer to keep the data separated for smokers and non-smoker (Table 3, former Table 5). In fact, some values were statistically different between the groups. Additionally, according to reviewer’s suggestion, we have added the values reported in literature for non-smoker environmental or occupationally exposed subjects. Moreover, we have added the following sentence:

Results – Line 196

 “In urine, biomarker concentrations were consistent with those reported in the literature for environmental or occupationally exposed non-smokers”.

  1. Not clear if this “smoker/non-smoker” status was based on cotinine cutoff of > 30 ug/L page 4 Line 121 or self-reported status as listed in Table 2 (described page 3 line 101-104).

REPLY: Please, see answer to 2.2. Question.

  1. In text you say lognormally distributed variables were reported as GM and GSD and that is shown in Table 3 and 4 but then you use pearson’s correlation to examine the relationship of biomarkers in SP and urine…..if these were on log values please specify, if not, they should be! The correlation of biomarkers of exposure in urine and SP and between each other can be discussed in text, I don’t really think you need tables as this is not surprising results (or could put in supplemental tables (Table 4, 7a, 7b) .

REPLY: correlation analyses were always performed on log values (log-log) as explained in the Statistical Analysis section. Tables 4, 7a and 7b have been moved to supplemental materials.

  1. Likewise, Figure 1 isn’t really necessary, since you have info in text.

REPLY: We agree with the reviewer. Figure 1 has been moved to supplemental material.

  1. It seems to me that the next logical step is the PCA as a way of dealing with this correlation (Table 8). Then the question becomes, do these Factor combinations relate to cotinine levels? So Table 6 could be modified to show that (missing VPmerc and AFIEM and no need for 8oxo data) OR better still, you could simply model the factors relative to Cotinine as outcome.  Reminder that in pg 4 line 139-141 you say urinary concentrations were expressed as a function of creatinine but here in Table 6 you list ug/l not ug/g creatinine. …… what was PCA done on?

REPLY: in SP the relationship  between smoking habits and factors was modest. Particularly, factor 1 was not significantly correlated with both smoking status (yes/no), while there was a significant but weak correlation between factor 2 and smoking status (R=0.29, p<0.01).

In urine, all three factors were related to smoking status, with a weak correlation for both factor 1 and 2 (R=0.20, p<0.05 and R=0.25, p<0.01, respectively) and high correlation for factor 3 (R=0.58, p<0.001).

By using cotinine values instead of smoking status (yes/no), results were overlapping, despite the limitations due to the bimodal distribution of cotinine values. In SP, the correlation between factor 2 and cotinine remained weak (r=0.32, p<0.001), while in urine there was only a strong correlation between factor 3 and cotinine (r=0.61, p<0.001).

Data indicate that the general contribution of recent smoking habits is clearly grouped when overall exposure was considered (urine), but not at the level of reproductive system (SP). These data should be confirmed measuring a higher number of cigarette smoke tracers.

To better clarify the concept, the following sentences have been added.

Results section at line 228.

“In SP, factor 1 was not significantly correlated with smoking status, while there was a significant but weak correlation between factor 2 and smoking status (R=0.29, p<0.01). In urine, all three factors were related to smoking habits, with a weak correlation for both factor 1 and 2 (R=0.20, p<0.05 and R=0.25, p<0.01, respectively) and high correlation for factor 3 (R=0.58, p<0.001). By using cotinine values instead of smoking status (yes/no), results were overlapping, despite the limitations due to the bimodal distribution of cotinine values. In fact, in SP, the correlation between factor 2 and cotinine remained weak (r=0.32, p=0.001), while in urine there was only a strong correlation between factor 3 and cotinine (r=0.61, p<0.001)”.

Some considerations are now present in line 387 and 394.

  1. Table 9. Is this on logs of outcomes and exposure biomarkers? If you want to do a uncontrolled analysis of outcomes with biomarkers…..why not use log of 8-oxo and the unlogged biomarker level as predictor….then Beta x100 is the percent change per unit increase in 8-oxo ?
    1. This table has no unites of 8-oxo
    2. Reminder that in pg 4 line 139-141 you say urinary concentrations were expressed as a function of creatinine but here in Table 9 you list ug/l not ug/g creatinine

REPLY: Thank you for this observation. There was a mistake because in this case the measurement units do not give beneficial information. Therefore, in accordance with other tables describing correlation results, measurement units are not reported.

  1. The Tables 10a, b are confusing….and what happened to 8-oxoGua?

REPLY: Regarding 8-oxoGua, multivariable analysis was not statistically significant for all of the three factors, underlining the absence of relationship between exposure and 8-oxoGua concentration, as biomarker of effect (data not shown). This result was reported in the test with the phrase: “while in the case of 8-oxoGua analysis was not statistically significant (data not showed).” However, we agree that this statement is not clear enough. Therefore, it was erased and the following sentence was added:

Results-Line 259

“Regarding 8-oxoGua, multivariable analysis did not show any statistically significant factor (data not shown).”

Discussion - Line 381

“The absence of relationship between exposure and 8-oxoGua concentration is probably due to the highest inter- and intra individual variability  of 8-oxo-Gua among biomarkers of nucleic acid oxidation investigated [27,31].”

  1. is the outcome logged or not?

REPLY: All statistical analyses were performed on log-values.

  1. No case is made for why you include sperm motility or volume SF (isn’t this quite variable per subject collection, not really a biological factor?)

REPLY: Based on the univariate correlations with biomarkers of effect, both sperm motility and semen volume were included as biological factors affecting the quality of sperm and chemical compound concentrations, respectively. In particular, a different semen volume may represent a different dilution factors for both spermatozoa and substances contained in it.

(…) or log creatinine (in pg 4 line 139-141 you say urinary concentrations were expressed as a function of creatinine).

REPLY: Creatinine was log-transformed to maintain the coherence with the way used to report the concentration of urinary biomarkers. The biomarkers of effect were always considered log-transformed. PCA factors were calculated by using the log-transformation of all the variables involved in it non normalized for creatinine, being the relationships in the same matrix.

  1. Why use smoker yes/no when you have cotinine which is a better marker of smoking status?

REPLY: Please, refer to answer to Question 10.

Editorial

Page 2. Line 53-54 add this to previous paragraph.

REPLY: Correction done.

Page 3 line 97 should be enrollment.

REPLY: Correction done.

Page 3 Line 98 specify organization that takes responsibility for ethical committee

REPLY: The District of Parma had only one Ethical Committee for all the subjects located in it until to 2016, namely local ethical committee. Now, the Ethical Committee groups different districts at the regional level (http://www.aou.mo.it/ComitatoEticoAVEN, only in Italian).

Page 3 Line 101 should be Table 2.

REPLY: Correction done.

Page 3 Line 115 write out seminal plasma (SP) and define it since you also use the terms semen under sample collection analysis section or at least be consistent on which you use.

REPLY: According to reviewer’s advice, “semen” and “seminal plasma” terms have been better clarify. The following sentences were added.

2.2. Sample Collection and Analysis, Line 118

“Within two hours from ejaculate collection and its liquefaction, semen was fractionated collecting the acellular component known as seminal plasma (SP). The cellular fraction, which in physiological conditions mainly corresponds to spermatozoa was not available in this study.”

Table 2 footnote What is DS? Should be Median.

REPLY: DS was replaced by Standard Deviation.

Table 3 What is footnote DS? Do you mean GM for Geometric Mean and GSD for Geometric Standard Deviation??

REPLY: There was a mistake. DS was replaced by Geometric Standard Deviation. Table 2 and Table 3 were fused together.

Page 6 line 185 divide the words were further

REPLY: Correction done.

Reviewer 2 Report

This is an interesting study that presents and discusses the relationship between low doses of exposure to environmental contaminants and oxidized products of nucleic acids in males attending a fertility center.

Although additional research is needed to confirm findings of this study, the study contributes to increase the scientific evidence on the vulnerability of the testicles to xenobiotic exposure even at low concentration.

Furthermore the data clearly demonstrate that Seminal Plasma (SP) and Urine provide different information and it is confirming the possibility of using SP to assess the exposure to reproductive toxicants both in environmental and occupational medicine.

Publication is recommended

No corrections needed

Author Response

This is an interesting study that presents and discusses the relationship between low doses of exposure to environmental contaminants and oxidized products of nucleic acids in males attending a fertility center.

Although additional research is needed to confirm findings of this study, the study contributes to increase the scientific evidence on the vulnerability of the testicles to xenobiotic exposure even at low concentration.

Furthermore, the data clearly demonstrate that Seminal Plasma (SP) and Urine provide different information and it is confirming the possibility of using SP to assess the exposure to reproductive toxicants both in environmental and occupational medicine.

Publication is recommended

No corrections needed

REPLY: we thank the referee for its encouraging comment.

Reviewer 3 Report

The submitted manuscript is a descriptive study of xenobiotic concentrations in the semen of males who were recruited from a fertility center. Study groups were selected based on the separation of patients who had known causes of infertility versus those who did not yet have a diagnosis. The latter group comprised the study population. The study population was subdivided into two groups: one group (IMI) exhibited altered sperm chrematistics; the other group (with no alterations in sperm characteristics) was considered the control group. It is not clear what causative information about infertility can be gleaned from two groups of infertile men without comparison to “normal” controls.

In Table 2, the authors compare the demographic and seminal characteristics of both groups. They report statistical differences between the groups based on percentage of motile sperm and percentage of abnormal sperm. However, the footnotes in the table indicate that the values presented are median values for the IMI group but are means for the controls. It is not clear how such different values can be compared statistically.

The authors expended a great deal of effort to measure concentrations of numerous xenobiotics and their respective metabolites in the seminal plasma and urine of both groups. There are two issues with the analyses of these data. The first issue relates to the inability to directly compare urine to semen because urine is the fluid that removes xenobiotic materials from the body (and often the materials have been metabolized and conjugated to make them more readily excreted), whereas semen is a fluid that will supply nutrition to ejaculated sperm. An explanation is required of why the comparisons between urinary and seminal concentrations are being made and how they should be interpreted.

The second issue relates to the authors’ repeated conclusions that finding the xenobiotic materials in the semen is an indicator of testicular exposure to and damage by these substances. Finding xenobiotics in the semen is not a marker of testicular exposure. The testis is well-protected from xenobiotics in the blood or tissue fluid by the testis-blood-barrier (TBB). The authors need to explain why they believe their findings support testicular degradation and how the xenobiotic materials arrived at the testis in sufficient concentrations to damage developing sperm. Further, they need to explain how the materials are concentrated in the testes. The reason for needing to explain concentration is that fluid from the testes makes up no more than 2-5% of seminal plasma volume (and most of that volume is due to the presence of the 200 -300 million sperm released in an ejaculate). The vast majority of the volume of seminal plasma comes from fluids released by the seminal vesicles (65=75%) and prostate gland (25-30%), with smaller contributions from the epididymis (~1%), and bulbourethral glands (<1%). Importantly, because most of the fluids in semen are released just before ejaculation, those fluids are in contact with sperm for only a few minutes within the vagina. Consequently, they cannot account for the presence of abnormal sperm.

The data contained in the manuscript may be useful as determinants of exposure to contaminants; however, the present manuscript does not clearly explain the authors’ hypothesis in the context of biological plausibility. The authors are encouraged to re-think the presentation of the study rationale and their interpretation and discussion of results.

Author Response

REVIEWER 3

The submitted manuscript is a descriptive study of xenobiotic concentrations in the semen of males who were recruited from a fertility center. Study groups were selected based on the separation of patients who had known causes of infertility versus those who did not yet have a diagnosis. The latter group comprised the study population. The study population was subdivided into two groups: one group (IMI) exhibited altered sperm chrematistics; the other group (with no alterations in sperm characteristics) was considered the control group.

  • It is not clear what causative information about infertility can be gleaned from two groups of infertile men without comparison to “normal” controls.

REPLY: We do agree with the referee. Our sentences in “Study Disegn” paragraph could give rise to misunderstandings. In fact, we classified one studied group as control owing to no alterations in sperm characteristics and since the cause of infertility was the female factor.

Therefore, the sentence “…owing to no alterations in sperm characteristics and was due to female counterpart” was added in 2.1. Study Design” line 96.

  • In Table 2, the authors compare the demographic and seminal characteristics of both groups. They report statistical differences between the groups based on percentage of motile sperm and percentage of abnormal sperm. However, the footnotes in the table indicate that the values presented are median values for the IMI group but are means for the controls. It is not clear how such different values can be compared statistically.

REPLY: Thank you for bringing that to our attention. In fact, there was a typo and all sperm characteristics (semen volume, sperm count, motility and morphology of sperm) are expressed as median (25th-75th percentile). The comparison was done with a non-parametric test (Mann-Whitney test).

  • The authors expended a great deal of effort to measure concentrations of numerous xenobiotics and their respective metabolites in the seminal plasma and urine of both groups. There are two issues with the analyses of these data.
    1. The first issue relates to the inability to directly compare urine to semen because urine is the fluid that removes xenobiotic materials from the body (and often the materials have been metabolized and conjugated to make them more readily excreted), whereas semen is a fluid that will supply nutrition to ejaculated sperm. An explanation is required of why the comparisons between urinary and seminal concentrations are being made and how they should be interpreted.
    2. The second issue relates to the authors’ repeated conclusions that finding the xenobiotic materials in the semen is an indicator of testicular exposure to and damage by these substances. Finding xenobiotics in the semen is not a marker of testicular exposure. The testis is well-protected from xenobiotics in the blood or tissue fluid by the testis-blood-barrier (TBB). The authors need to explain why they believe their findings support testicular degradation and how the xenobiotic materials arrived at the testis in sufficient concentrations to damage developing sperm. Further, they need to explain how the materials are concentrated in the testes. The reason for needing to explain concentration is that fluid from the testes makes up no more than 2-5% of seminal plasma volume (and most of that volume is due to the presence of the 200 -300 million sperm released in an ejaculate). The vast majority of the volume of seminal plasma comes from fluids released by the seminal vesicles (65=75%) and prostate gland (25-30%), with smaller contributions from the epididymis (~1%), and bulbourethral glands (<1%). Importantly, because most of the fluids in semen are released just before ejaculation, those fluids are in contact with sperm for only a few minutes within the vagina. Consequently, they cannot account for the presence of abnormal sperm.

The data contained in the manuscript may be useful as determinants of exposure to contaminants; however, the present manuscript does not clearly explain the authors’ hypothesis in the context of biological plausibility. The authors are encouraged to re-think the presentation of the study rationale and their interpretation and discussion of results.

REPLY: We agree with the reviewer.

Looking at point (a): Urine is the biological matrix commonly used to do the biological monitoring of the xenobiotics considered in this study and it is representative of overall exposure, without giving no information about specific organs, that may be a target. As specified below, seminal plasma collected in a fertility center, although of mixed origin, can be considered as representative of reproductive system. Therefore, a direct comparison is fundamental to understand whether urine may be representative of reproductive system and therefore is the “gold standard” to perform biological monitoring in subjects exposed to compounds with reproductive toxicity in men.

Some sentences are inaccurate and could give rise to misunderstanding.  Firstly, “testicular exposure” is inappropriate and must be replaced by “male reproductive tract exposure”. Moreover, we think that a brief rational about biological monitoring of urine together with the description of SP origin could increase the understanding of the work.

Finally, unlike in vivo, where seminal plasma is in contact with sperm for only a few minutes, in our study the semen was collected and separated within 2 hours with a potential effect of SP toward spermatozoa or vice versa. Therefore, a quantification of oxidative stress directly on spermatozoa would increase our knowledge about the interactions between the cell and seminal plasma. Unfortunately, it was not possible because spermatozoa were unavailable for almost all subjects.

Therefore, the following sentences have been added.

Discussion-line 278

“Biological monitoring is widely applied in environmental/occupational toxicology due to its remarkable utility on providing an efficient and effective means of measuring human exposure to chemical substances. It is aimed at defining the existence of a former exposure and quantifying the level of internal dose [33,51]. Blood and urine are by far the most used matrices, as they provide pertinent information about the biotransformation of xenobiotics in the body. In the study, we focused our attention on urine analysis, because the exposure evaluation to the substances selected in this study are generally performed on urine. On the other hand, SP analysis may provide specific information on the male reproductive system and could not reflect the overall exposure [52]. Several studies have highlighted that oxidative stress in the male reproductive tract is related to toxic agent exposure and it is implicated in low sperm quality (motility, viability and increasing sperm morphological defects) [6,7,9,11,12]. In fact, compared with other cells, human spermatozoa are extremely vulnerable to oxidative stress because of their high levels of polyunsaturated fatty acids and the little cytoplasm sequestering defensive enzymes [8,10,18]. A higher xenobiotic exposure (e.g. acute exposure) could also impair sperm membrane and affect spermatozoa triggering a cascade effect (e.g.  increased cell permeability, enzyme inactivation, protein oxidations and DNA damage). Unfortunately, spermatozoa were not available in this study. Therefore, to have information about male reproductive tract we focused our attention on SP analysis. In fact, several secreting epithelia and glands differentially concur to the production of SP. In particular, vesicular glands and prostate provide the main contribution releasing about 65% and 25% of SP, respectively, while testes and epididymes supply solely 2–5% of SP. As a mixture of various secretions from male accessory sex glands contains several biochemical compounds also deriving from chemical exposure [53].”

2.2. Sample Collection and Analysis, Line 119

“Within two hours from ejaculate collection and its liquefaction, semen was fractionated collecting the acellular component known as seminal plasma (SP). The cellular fraction, which in physiological conditions mainly corresponds to spermatozoa was not available in this study.”

Round 2

Reviewer 1 Report

I believe the manuscript has been significantly improved and now warrants publication in IJERPH Regards

Author Response

We thank the referee for his/her encouraging comment.

Reviewer 3 Report

This revision has considerably improved the manuscript. Many of my previous comments have been successfully addressed. There are, however, some remaining issues.

The abstract has not been changed sufficiently to match up with newly added text. Consequently, the abstract does not parallel the information presented in the body of the manuscript. As an example, at line 35 the statement is made that the study demonstrates the testicles are a target organ for environmental pollutants. Most to the text now uses “male reproductive tract,” which is more accurate.

The authors have left in their paper claims of DNA damage to sperm (e.g. line 66 as well as at other places). It is not clear how this would happen if the spermatozoa are the target. The DNA within sperm is in a condensed form of heterochromatin which is not readily available for attack by pollutants in the seminal plasma. Given that fact and the relatively short period that the sperm were in contact with seminal plasma, the idea put forth that 8-oxo-dGuo is a result of repair enzymes (line 77) seems rather weak. Could the free purines and ribonucleosides, which are found in the blood (where they exist at~0.4-0.6 μM [e.g., Traut. 1994. Mol Cell Biochem 140:1-22]) have been incorporated into the secretions of the seminal vesicles and/or prostate? If so, could this be the source of be the source of 8-oxo-dGuo that was found in the seminal plasma?

Alternatively, could the pollutants have crossed the blood-testis barrier (BTB) and affected the actively dividing spermatogonia resulting in damaged DNA? The plausibility of this concept must be considered in conjunction with (1) the arterial blood flow to the testis is rather low (about 9.6 mL/min out of a cardiac output of ~4.7 L/min) and (2) the BTB is even more difficult to cross than the blood brain barrier. Regardless, the authors’ thought on the two preceding topics would be welcome additions to their manuscript.

In the caption for Table 2, please note that the abbreviation “N.S.” does not appear in the table and so it should be deleted from the caption.

Author Response

REVIEWER 3

This revision has considerably improved the manuscript. Many of my previous comments have been successfully addressed. There are, however, some remaining issues.

The abstract has not been changed sufficiently to match up with newly added text. Consequently, the abstract does not parallel the information presented in the body of the manuscript. As an example, at line 35 the statement is made that the study demonstrates the testicles are a target organ for environmental pollutants. Most to the text now uses “male reproductive tract,” which is more accurate.

We agree with the reviewer and “testicles” has been replaced by “reproductive tract itself”. 

The authors have left in their paper claims of DNA damage to sperm (e.g. line 66 as well as at other places). It is not clear how this would happen if the spermatozoa are the target. The DNA within sperm is in a condensed form of heterochromatin which is not readily available for attack by pollutants in the seminal plasma. Given that fact and the relatively short period that the sperm were in contact with seminal plasma, the idea put forth that 8-oxo-dGuo is a result of repair enzymes (line 77) seems rather weak. Could the free purines and ribonucleosides, which are found in the blood (where they exist at~0.4-0.6 μM [e.g., Traut. 1994. Mol Cell Biochem 140:1-22]) have been incorporated into the secretions of the seminal vesicles and/or prostate? If so, could this be the source of be the source of 8-oxo-dGuo that was found in the seminal plasma?

Alternatively, could the pollutants have crossed the blood-testis barrier (BTB) and affected the actively dividing spermatogonia resulting in damaged DNA? The plausibility of this concept must be considered in conjunction with (1) the arterial blood flow to the testis is rather low (about 9.6 mL/min out of a cardiac output of ~4.7 L/min) and (2) the BTB is even more difficult to cross than the blood brain barrier. Regardless, the authors’ thought on the two preceding topics would be welcome additions to their manuscript.

We agree with the reviewer that the sentences related to DNA damage sperm could cause misunderstanding, especially since we focused our work on “male reproductive tract” generally. Therefore, the following sentences have been deleted:

Introduction – Line 65

“Increased levels of seminal oxidative stress is related to sperm dysfunction through different mechanisms that include lipid peroxidation of sperm plasma membrane and impairment of sperm metabolism, motility, and fertilizing capacity [19,20] together with DNA damage [21,22].”

Discussion – Line 290

In fact, compared with other cells, human spermatozoa are extremely vulnerable to oxidative stress because of their high levels of polyunsaturated fatty acids and the little cytoplasm sequestering defensive enzymes [8,10,18].

Discussion – Line 380

“In fact, spermatozoa are significantly prone to oxidative damage particularly to the cell membrane and the DNA [61].”

At the same time, we believe it is useful to add the reviewer’s suggestions relating the hypothesis on the origin of oxidized products of nucleic acids in SP.

Discussion – Line 289

“In this case, the DNA damage to sperm due to xenobiotics in SP from widespread pollution, is difficult to explain. In fact, the low exposure and the relatively short period of contact between sperm and SP (< 2 hours) makes it unlikely that oxidized products of nucleic acids are the result of repair enzyme activity. Additionally, the DNA within sperm is in a condensed form of heterochromatin which is not readily available for attack by pollutants in SP. Alternatively, the pollutants could have crossed the blood-testis barrier (BTB) and affected the actively dividing spermatogonia resulting in damaged DNA. The plausibility of this concept is low in conjunction with the fact that (1) the arterial blood flow to the testis is rather low (about 9.6 mL/min out of a cardiac output of ~4.7 L/min) [49,50] and (2) the BTB is one of the tightest blood-tissue barriers known to exist in mammalian tissues, even more difficult to cross than the blood brain barrier [51-53]. In fact, one of the main role of the BTB is to prevent small molecules, and hence also harmful contaminants, from passing through the paracellular space [54]. On the other hand, despite being one of the strongest blood-tissue barriers, recent studies reported adverse effects of xenobiotics, such as bisphenol A and monobutyl phthalate, on Sertoli cell BTB [51,52]. To clarify this aspect, further investigation should be performed particularly on spermatozoa, that unfortunately were not available to this study.”

Discussion – Line 308

Additionally, free purines and ribonucleosides in blood (0.4-0.6 μM concentration) [56], after being incorporated into the secretions of the seminal vesicles and/or prostate, could become a possible target of oxidative stress in SP.”

New References

[49] Tarhan et al, 2003

[50] Goktas  et al, 2017

[51] Tao et al, 2019

[52] de Freitas et al, 2016

[53] Su et al, 2011

[54] (Yan and Cheng, 2005).

[56] Traut, 1994

In the caption for Table 2, please note that the abbreviation “N.S.” does not appear in the table and so it should be deleted from the caption.

We want to thank the reviewer for this thorough draft control. The caption of the Tables have been carefully revised and all typo errors corrected.

attached file.

Round 3

Reviewer 3 Report

The revision has responded to my comments and has improved the manuscript.  I am satisfied.